**Data Availability Statement:** All relevant data are within the manuscript and its Supporting Information files. I have attached a code sheet and

# Perceptions and practices of imaging personnel and physicians regarding the use of brain MRI for dementia diagnosis in Uganda

Rita Nassanga[1]*, Noeline Nakasujja[2], Mark Kaddumukasa[3], Stephen E. Jones[4], Martha Sajatovic[5], Michael Grace Kawooya[6]

1 Department of Radiology, College of Health Sciences, Makerere University, Kampala, Uganda,
2 Department of Psychiatry, College of Health Sciences, Makerere University, Kampala, Uganda,
3 Department of Medicine, College of Health Sciences, Makerere University, Kampala, Uganda, 4 Imaging Institute, Cleveland Clinic Lerner School of Medicine, Cleveland, OH, United States of America, 5 Case Western Reserve University, Cleveland, OH, United States of America, 6 Ernest Cook Ultrasound Research and Education Institute (ECUREI), Kampala, Uganda

* ritanassanga@gmail.com

## Abstract

### Introduction

Diagnosing dementia remains challenging in low-income settings due to limited diagnostic options and the absence of definitive biomarkers. The use of brain MRI in the diagnosis of dementia is infrequent in Uganda, and even when it is used, subtle findings like mild regional atrophy are often overlooked, despite being crucial for imaging diagnosis.

### Objective

The purpose of this study was to explore the perceptions and practices of imaging personnel and physicians regarding the use of brain MRI as a diagnostic approach for dementia in Uganda.

### Methods

This was an exploratory qualitative study involving radiologists, technologists, senior house officers and psychiatrists. The participants were 25 in total. Data was collected through key informant interviews and focus group discussions and analyzed thematically using an inductive approach.

### Results

The study revealed three key themes: Brain MRI Practices for Diagnosing Dementia, Facilitators of Appropriate MRI Use, and Barriers to Appropriate Use of Brain MRI. Sub-themes under these themes included cost considerations, poor and good MRI practices, MRI as a standard operating procedure, positive attitudes towards brain MRI, and barriers such as structural, financial, operational, technical, and patient-related issues. Participants acknowledged the high accuracy and superiority of brain MRI for diagnosing dementia and

how the codes were categorized. The quotes however are not provided because they contain identifying information and would require authorization from the ethics board to access them. Any requests or questions about data can be made through the School of Medicine Research and Ethics Committee (SOMREC) via uncstresearch@uncst.go.ug or the REC admin at aidan.kiseka@gmail.com.

**Funding:** The funder of the study is National Institute of Neurological Disorders and Stroke, grant number is D43TW010132 and the Grant Recipient is Prof Mark Kaddumukasa.The funders had no role in study design, data collection and analysis, decision to publish, or preparation of the manuscript.

**Competing interests:** The authors have declared that no competing interests exist

recognized it as the standard of care. However, its use in Uganda is limited due to high costs, restricted access, mechanical failures, patient claustrophobia, myths and misconceptions, and interpretation difficulties by radiologists and inappropriate protocols by technologists.

## Conclusion

The study identifies barriers to effective brain MRI use for dementia diagnosis in Uganda, including limited training, high costs, and uneven equipment distribution. Despite this, providers are positive about MRI adoption. Enhancing training, awareness, and phased rollouts can improve outcomes. Future research should focus on similar low-resource settings for validation.

## Introduction

Dementia refers to a spectrum of diseases that affect the brain causing cognitive and behavioral disturbances with associated disruption of activities of daily living [1]. It affects 7–8% of individuals older than 65 years old and 30% of individuals older than 80 years old [2]. Dementia, particularly Alzheimer's disease (AD), stands as the most extensively researched and prevalent neurodegenerative condition. Alzheimer's disease (AD), along with frontotemporal dementia (FTD), vascular dementia, and dementia with Lewy bodies (DLB), collectively encompass 85% of degenerative dementia cases [3].

The global prevalence of dementia was estimated to be as high as 24 million in 2011, 44 million in 2014, and it is expected to double every 20 years through to 2040 or even triple by 2050 [4, 5]. Dementia prevalence ranges from 2.3% to 20.0%, and the incidence rate is 13.3 per 1,000 person-years, with rising mortality rates in areas of Africa undergoing rapid change. More than half of dementia cases occur in Low and Middle Income Countries (LMICs) [6] posing a significant challenge to modern healthcare systems. The growing prevalence of dementia in these regions is a major concern due to its substantial socio-economic burden, negatively impacting patients' quality of life, straining families and healthcare resources, and necessitating urgent improvements in diagnosis and cares [7, 8]. The main driver of dementia figures is age and thus the anticipated ageing of the population and rapid demographic transitions with increased life expectancy in Africa and Sub-Saharan African (SSA) may translate in a higher prevalence and absolute number of people living with dementia as observed in other developing regions [9].

In Uganda, a nation grappling with economic constraints, life expectancy has shown a notable upward trajectory, progressing from 39.0 years in 1951 to 46.2 years in 2000, and further advancing to 63.4 years in 2020, with a subsequent increase to 64.7 years by 2024 [9] thus increasing the population of the elderly.

The standard of diagnosing dementia involves biomarkers such as blood, cerebrospinal fluid, and neuroimaging [10] however, it is very challenging to diagnose especially in LMICs because of limited work up options leaving the diagnosis to majorly clinical history, cognitive tests and neuroimaging. MRI offers high-spatial-resolution and has evolved to detect subtle morphological changes early on, aiding in the identification of neurodegenerative processes with high accuracy. This advanced imaging not only assists in surgical treatment for many dementia patients but also helps rule out treatable or reversible causes of the condition [1, 11–13].

Model paper [14] In a previous study on perceptions on dementia diagnosis, it was reported that while MRI is recommended for diagnosis, there is limited skilled workforce [14] and thus estimates suggest that a significant proportion, ranging from 62% to 90%, of individuals with dementia in developing countries are likely not diagnosed [15, 16]. In Africa, health professionals are skeptical to use it, and there is a significant lack of awareness about diagnostic tools like MRI, [17], while in Uganda, the high prevalence of dementia is exacerbated by limited use of advanced diagnostic tools like MRI due to resource constraints as well as key gaps in the knowledge and skills of healthcare workers in dementia assessment and diagnosis [13, 18].

The use of brain MRI in the diagnosis of dementia is infrequent in Uganda [13, 19], and even when it is used, subtle findings like mild regional atrophy are often overlooked, despite being crucial for imaging diagnosis Among the seven imaging centers in Uganda's capital, only one employs a standardized protocol for brain MRI examinations for dementia. Furthermore, there is a lack of uniform guidelines for interpreting brain MRI results, which would provide radiologists with essential checkpoints for diagnosing dementia. Limited published data exist on the use of brain MRI for dementia diagnosis in low-resource settings, where demographic profiles may differ from those in high-income countries. This study aimed to explore the perceptions and practices of imaging personnel and physicians regarding the use of brain MRI in diagnosing dementia. In the context of the study, practices referred to uptake or utilization of MRI.

## Methods

### Study design

This was an exploratory qualitative study conducted in health facilities with MRIs.

### Study setting

The study took place at seven urban based health facilities in Kampala, Uganda, and each had a 1.5T MRI scanner. In each of these centres, various MRI investigations take place including assessment of patients with dementia.

### Recruitment period

Recruitment of participants started on the 10th. January 2024 and ended on the 01st. March. 2024.

### Participants and sampling

This study involved 25 participants with knowledge on imaging and diagnosis of dementia. The participants comprised three radiologists, six Radiology residents, eight Psychiatry residents, two psychiatrists, two neurologists, and four imaging technologists from six different health facilities. This being a purely qualitative study, the participants were purposively selected on basis of their experience with brain MRI or diagnosis of dementia and thus were more suitable to describe their practices and perceptions to the researcher. They were sampled to the point of saturation.

### Inclusion criteria

The participants included in the study had to be established experts in dementia diagnosis and care in any of the following disciplines: medicine, psychiatry, radiology, neurology, neuropsychology, and psychiatry as well as currently involved in clinical practice or residency training and whose training involves aspects of dementia management.

## Exclusion criteria

The exclusion was non-availability of participants for FGDs or KIIs.

## Data collection

Data was collected using Focus Group Discussions (FGDs) and Key Informant Interviews (KIIs). The interview guides were designed to capture a broad overview of current diagnostic behaviours as well as practices and perceptions associated with diagnosis of dementia. The participants were contacted by email and phone call. A total of 30 experts were contacted and 25 responded and agreed to participate. The interviews were conducted between February and April 2024.

**Focus group discussions.**   Two FGDs were conducted, one with eight psychiatry residents and the second with six Radiology residents. Two FGDs were conducted because these are the two groups of residents that deal with dementia patients. These were conducted by two experienced research assistants (one moderated the session while the other took notes) from a quiet place at one of the health facilities in the study. The FGDs were conducted using a semi-structured guide exploring participants' practices and perceptions of using brain MRI for dementia diagnosis. The FGDs were conducted in English, audio-recorded and later transcribed verbatim for analysis. Key notes were also taken during the sessions to capture participants' expressions during the discussions. This was done to ensure that vital information is not lost or missed during analysis. The questions for the discussions were informed by literature and first pre-tested with four experts to ensure their credibility.

**Key informant interviews.**   Eleven face-to-face KIIs were conducted with senior consultant neurologists, radiologists, technologists and psychiatrists using a semi-structured interview guide to explore their practices, perceptions, feasibility and acceptability of using brain MRI for diagnosing dementia. The KIIs were conducted in English, audio-recorded and later transcribed verbatim for analysis. They were conducted to a point where no new information was emerging. Saturation was achieved at the 10[th] participant but we included an extra interview to confirm saturation. The interview guide questions were also informed by literature and were first pre-tested with two experts to ensure their credibility.

## Data management and analysis

All responses from KIIs and FGDs were audio-recorded and transcribed verbatim in English. The transcripts were cross checked with the recordings and field notes for accuracy. The data was stored on a password- protected computer owned by the researcher and thereafter imported to ATLAS ti Version 23 for systematic data management and analysis. Data was analyzed thematically following four steps as suggested by Malterud, 2012 [20]. Each transcript was read twice to get an overall impression of the data and to develop initial codes. A coding dictionary with mutually exclusive code definitions was then developed and was refined often in the course of the analysis. The codes were assigned inductively to different chunks of text to capture phenomena in the participants' perspective. The codes were then categorized into subthemes and major themes. To improve the rigor, three participants of the FGDs were invited and presented with preliminary subthemes before generation of the major themes exploring the practice and perceptions of using brain MRI in the diagnosis of dementia.

## Rigor and trustiworthiness

Rigor was ensured in this study through a number of ways. Confirmability was ensured through documenting the steps taken to conduct the study while dependability was ensured by

ensuring that the KIIs and FGDs probed all issues and two methods of data collection were used for triangulation. Transferability was ensured by documenting all steps taken to conduct the study so that it can be replicated in other areas following the steps.

## Ethical considerations

This study was approved by the Makerere University School of Medicine Research Ethics Committee (Protocol Number: Mak-SOMREC-2022-337). Written informed consent was obtained from each of the participants prior to data collection and participants were not identified by their responses. All participant responses were kept confidential and only the researchers had access to the raw data.

## Results

### Demographic characteristics

The average age of the participants was 36 years (range, 27–73 years) and all had a university degree or higher. Six of the participants were female while 19 were male. The sample included ten psychiatrists, two neurologists, three radiologists, and ten radiographers, and their average experience in imaging or dementia diagnosis and management was 7 years (range, 2–40 years).

Analysis of the responses yielded three overarching themes including; 1) Brain MRI practice in the diagnosis of dementia, 2) Facilitators to appropriate use of brain MRI in dementia and 3) Barriers to appropriate use of brain MRI to diagnose dementia. These are summarized in the Table 1 below along with their respective sub-themes:

The above themes and and sub-themes are further described below:

### Theme A: Brain MRI practice in the diagnosis of dementia

**Sub-theme 1: Consideration of the cost implication to the patient.** All participants showed a unanimous view regarding the superiority of brain MRI over other imaging modalities for diagnosing dementia. This can be seen in the response below:

*"[. . .] Of course, there are many conditions that influence this decision. The ideal would be an MRI but if a patient can't afford it, we can look at the first-line alternative which is the CT scan to try and analyze what could be the possible cause."–Neurologist, 10 years of experience*

However, many highlighted the importance of considering the financial resources and implications for the client before recommending an MRI. In cases where it is not feasible, the participants reported resorting to alternatives such as the CT scan or relying on clinical diagnosis. The aspect of financial considerations can be seen to sweep through the responses below:

*"[. . .] There are people who come and can't afford the MRI so sometimes the doctor may be compelled to change to CT to at least get a picture of what's happening in the brain."—Radiographer, 7 years of experience.*

*"[. . .] As per radiology, mainly we use MRI however as I said, it depends on the interest of the doctor or the person referring because there are people who refer with the interest of looking at the cost implications. For example, because MRI is expensive, they go for other cheaper tests like a CT scan."- Radiologist, 14 years of experience.*

**Table 1. Themes of brain MRI practice and perceptions with illustrative quotes.**

| MAIN THEME | SUBTHEME | SAMPLE CODES | *ILLUSTRATIVE QUOTES* |
|---|---|---|---|
| Brain MRI practice in the diagnosis of dementia | Circumstances under which brain MRI is recommended or used | • Circumstances for MRI—affordability to the patient<br>• Circumstances for MRI—physician interest<br>• Circumstances for MRI—to rule out other causes | *"Of course, there are many conditions that influence this decision. The ideal would be an MRI but if a patient can't afford it, we can look at the first-line alternative which is the CT scan to try and analyze what could be the possible cause."* |
| | Good MRI practice | • DM diagnosis—MRI<br>• MRI protocols used<br>• MRI referral | *"[. . .] I always send them (patients) to imaging centers in town for example Kampala Imaging Center or Nsambya Hospital. They exclude other possible causes however, for Alzheimer's dementia, you have to have a certain specificity in reporting. . . I don't use other imaging modalities because I don't think they are informative enough."* |
| | Poor MRI practice for dementia | • Few DM-MRI patients seen<br>• DM diagnosis—Clinical<br>• DM diagnosis—CT | *,"[. . .] we see like 1 to 2 patients a month. . .I think that not all patients are brought here. . . people don't want to come or bring their patients to mental hospitals when they are not clearly "mad" as they describe it in the general population."* |
| Enhances early detection | • Higher specificity<br>• Higher sensitivity | • Perc. benefit of MRI-more specific<br>• Perc. benefit of MRI-enhances early detection<br>• Perc. benefit of MRI-better/detailed images<br>• Perc. benefit of MRI-more sensitive | *"I am going to give an example of a patient that I actually got to diagnose with dementia through imaging at Nsambya Hospital; this patient had a normal CT scan but they were only able to see changes through MRI imaging. So, if we are to compare CT scans and MRI, with that experience that I had, it is better to use the MRI to be able to see these changes easily."* |
| Perceived barriers | • High cost<br>• Mechanical breakdowns<br>• Interpretation difficulties | • MRI experience—procurment delays<br>• MRI experience—expensive for the clients<br>• MRI experience—expensive for the facility<br>• MRI experience—mechanical breakdowns<br>• MRI experience—problem interpreting reports | *"[. . .] If I have 4 patients, only 2 might come for the test and the rest will say that the cost is too high. It is not that the patients don't want to have the test, it is because it is very costly."*<br>*"[. . .] It has broken down once or twice. It needed a spare part and we had to wait for it to arrive, however, it didn't take as long as expected because it only took a month or 2 weeks."* |
| Feasibility and adoptability | • Inaccessibility to MRI services<br>• Specialists gap<br>• Operational costs | • MRI nonfeasibility—knowledge gap of physicians<br>• MRI nonfeasibility—few MRI centers<br>• MRI nonfeasibility—long-term mechanical breakdowns<br>• MRI nonfeasibility—expensive to set up<br>• MRI nonfeasibility—unstable power | *"[. . .] For now, it is not so feasible because of all the challenges we have mentioned and we only have MRI in the central region yet most of the population is in other areas of the country. I am more certain that there are more elder people with these conditions, but given that they don't even know where MRIs are in the country, they may not even request it."* |

**Sub-theme 2: Good MRI practice.** Neurologists and psychiatrists stated that they often recommend a brain MRI to confirm dementia following a clinical assessment as can be seen below:

*"[. . .] We are lucky that we have specialists here so people who have such symptoms are sent to neurologists. So, we diagnose it at the MRI and we are lucky that we have the MRI and we see the patients. Clinically, the doctor suspects but the MRI diagnoses."*–Radiologist, 15 years of experience

On the other hand, radiographers and SHOs mentioned different MRI protocols and sequences they use in their daily practice. This good MRI practice among radiographers was

facilitated by their ease in using MRI protocols and sequences and experiencing fewer mechanical breakdowns of the MRI machine in their facilities.

> *"[. . .] Regarding imaging for dementia, we use T1W, T2W, FLAIR, and diffusion-weighted imaging, and we also do the T1 IMPRAGE and CONTRAST T1W."*–FGD with Radiology SHOs, Mak CHS

Among the psychiatrists, the good practice was attributed to the steadily increasing number of MRI centers in the capital, Kampala.

> *"[. . .] I always send patients to imaging centers in town. They exclude other possible causes however, for Alzheimer's dementia, you have to have a certain specificity in reporting. . . I don't use other imaging modalities because I don't think they are informative enough."*- Psychiatrist, 40 years of experience

**Sub-theme 3: Poor MRI practice for dementia.** Most of the participants emphasized the rarity of using brain MRI to diagnose dementia, and this was mainly attributed to the high cost to the patient. Some emphasized that brain MRI is usually used as the last choice after trying all the cheaper options, such as clinical assessment and the CT scan.

Some key informants also attributed the poor MRI practice for dementia to the low numbers of patients with dementia and the prohibitive context of the health facility related to stigma. For instance, one radiologist in a mental facility stated:

> *"[. . .] we see like 1 to 2 patients a month. . .I think that not all patients are brought here. . . people don't want to come or bring their patients to mental hospitals when they are not clearly "mad" as they describe it in the general population."*

Another radiologist responded:

> *"[. . .] (we see) 1 to 2 cases a week. . . It (the number) is significant given that not everybody comes here due to some limitations. But sometimes we see no one with dementia or just 3 to 4 cases a month"*–Radiologist, 30 years of experience

## Theme B: Facilitators of appropriate use of brain MRI in the diagnosis of dementia

**Sub-theme 1: MRI in the Standard Operating Procedure (SOP) for dementia diagnosis.** Some participants stated that brain MRI is already in the SOP for diagnosing dementia in their facilities. However, this stood out more among participants in profit-for-profit compared to government health facilities. Neurologists and psychiatrists stated that they often recommend a brain MRI to confirm dementia following a clinical assessment. Radiographers demonstrated varied MRI protocols and sequences in their daily practice, facilitated by their proficiency in protocol utilization and encountering fewer mechanical breakdowns of MRI machines. Conversely, psychiatrists noted improved practice standards, credited to the rising number of MRI centers in the capital, Kampala. The following quotes from the participants further explain this theme regarding the facilitators and adoptability of diagnosing dementia in their health facilities:

> *"[. . .] if you look at our standard operating procedure, it is under MRI. It would be a confirmation of what we think would be the right thing for us to take."*–Radiographer, 13 years of experience

*"[. . .] Here, patients with certain symptoms are sent to specific doctors or they come to see a specialist who will know what investigation to ask for. What has made it easy is that there is good flow and the people are sort of directed to specialists. The thing is that equipment is available and running and most times they don't break down. Even when they break down, it doesn't take long and the specialists are available here as you have seen how busy we are"*– Radiologist, 15 years of experience

*"[. . .] I always send them (patients) to imaging centers in town for example Kampala Imaging Center or Nsambya Hospital. They exclude other possible causes however, for Alzheimer's dementia, you have to have a certain specificity in reporting. . . I don't use other imaging modalities because I don't think they are informative enough."*- Psychiatrist, 40 years of experience

**Sub-theme 2: Positive attitude toward using brain MRI for diagnosing dementia higher sensitivity.**   Many participants recognized that brain MRI is adept at detecting both subtle and structural alterations within the brain sometimes even before clinical manifestations, a capability not shared by other imaging modalities like CT scans. Ultimately, it allows early intervention or provides an opportunity to slow progression of disease. Here are a few excerpts from the responses:

*"[. . .] for someone to have a clinical manifestation of the disease, that disease could have reached a given stage. With imaging in this case MRI where we are sure all details will be picked, there are some structural or signal changes that may be picked that have not been manifested or given a clinical correlation yet. MRI helps in those aspects."*–FGD with Radiology SHOs, Mak CHS.

*"[. . .] I am going to give an example of a patient that I actually got to diagnose with dementia through imaging at Nsambya Hospital; this patient had a normal CT scan but they were only able to see changes through MRI imaging. So, if we are to compare CT scans and MRI, with that experience that I had, it is better to use the MRI to be able to see these changes easily."*– FGD with radiology SHOs Mak CHS

**Sub-theme 3: Higher specificity.**   The participants acknowledged that dementia is a very specific disease and commended the MRI for producing better and more detailed images of specific areas of the brain compared to other imaging modalities. Some neurologists emphasized MRI's advantage in cases where patients present with symptoms similar to dementia, such as those with diabetes or a history of trauma. Additionally, psychiatrists emphasized the importance of MRI in early diagnosis, particularly for individuals with vascular dementia. Many concurred that in diagnosing dementia, it would be a waste of the patient's resources sending them for a CT scan because it is less informative. Below were some excerpts from their responses:

*"[. . .] for anybody with memory cognitive impairment or poor scores with other co-morbidities like diabetes, and patients that have had different spots of trauma, an MRI is the priority. Initially, we would send them for a CT scan which would always come out normal and then we have to request for an MRI later. So, to me those above 55 to 60 years having issues with forgetfulness, the MRI is better because it is able to pick up those visions or whiteness around the ventricles or see any micro infarcts within the brain. So, it is much easier to know more diagnoses which will help you get a diagnosis faster than getting a series of tests that will even cost a lot of money."*–Neurologist, 10 years of experience

*"Comparing MRI with other modern tests, of course here we have CT scans and MRI but CT won't give us as many details as MRI would.. . . In CT scans, even if you give contrast, you may not see any difference within the brain or tissue but with MRI, the different sequences will help you see any signal changes in any particular area."*–FGD with radiology SHOs Mak CHS

### Theme C: Barriers to appropriate use of brain MRI to diagnose dementia

**Sub-theme 1: Structural and financial barriers.** *Inadequate MRI centers.* Participants reported that despite the limited number of MRI centers in Uganda, many of these are situated in the capital city, Kampala. This scarcity was attributed to the high costs associated with establishing and maintaining MRI facilities. Moreover, some participants disclosed having ongoing debt financing for the initial setup of their MRI centers. Consequently, the high cost of service provision is passed on to patients seeking these services, the majority of whom are not covered by insurance, significantly impacting the feasibility of brain MRI for dementia diagnosis. Below are excerpts from participants illustrating these challenges:

*"[. . .] For now, it is not so feasible because of all the challenges we have mentioned and we only have MRI in the central region yet most of the population is in other areas of the country. I am more certain that there are older people with these conditions, but given that they don't even know where MRIs are in the country, they may not even request it."*–FGD with psychiatry SHOs, Mulago Hospital

*"[. . .] Right now, government facilities have one MRI in Mulago but if they were available everywhere in the country, then it would be very easy to diagnose and the policymakers will be able to adapt. As I have told you, it is possible and it should be done because other countries have adapted it to diagnose dementia. I don't know why Uganda has to lag behind simply because we lack equipment."*–FGD with Radiology SHOs, Mulago Hospital

*Specialists' gap.* The participants pointed out the scarcity of specialized professionals in the country capable of generating and interpreting MRI reports, thereby complicating the feasibility and acceptance of this diagnostic approach. Moreover, some psychiatrists complained about frequently receiving inconclusive MRI reports from radiologists. They highlighted the novelty of brain MRI in Uganda, emphasizing that only a limited number of seasoned radiologists are centralized in the capital city, Kampala. Below are some quotes from the participants' responses:

*"[. . .] Right now, dementia diagnosis is done by specialists and very few make the diagnosis. As of now, specialists are in regional referral or national referral hospitals so, by trying to democratize MRI, we need to have specialists at a level where we can send patients for MRI and they can interpret them. This means that in every region we need to have radiologists and neurologists, which is extreme and expensive and can't be done in a short time, so it needs years of planning.*–FGD with radiology SHOs, Mak CHS

*"[. . .] In our low-income setting, it will mean that we have to train a number of radiologists and place them in different facilities which I think would be a huge cost. In our setting, I don't think it will be feasible. We also have to bear in mind that in the undergraduate training here, we have not been having an MRI and by the time you finish medical school with your first degree, you know little about how to interpret an MRI. So, if you make it a gold standard when the Medical officers don't know how to interpret images it would not be something that they would go for"*–FGD with psychiatrists, Mulago Hospital

*High cost*. Almost all participants acknowledged that the brain MRI scan is expensive for the majority of patients, except for a few key informants in private-for-profit facilities who stated that the majority of their clientele can afford the MRI scan. They emphasized that the cost is so high for patients that it delays dementia diagnosis despite subsidization of the service in certain public facilities. Some key informants also mentioned that the high cost discourages them from recommending an MRI and instead opt for a cheaper CT scan or a clinical diagnosis. Below are some quotes from the participants:

*"[. . .] Of course, cost is a very big issue. There are people who come and can't afford the MRI so sometimes the doctor may be compelled to change to CT to at least get a picture of what's happening in the brain. Some even have to change the appointment to another time or delay an examination because they can't afford the money for the scan and this can also delay the diagnosis."*–Radiographer, 7 years of experience

*"[. . .] The cost is a lot. Of course, some would be able to pay for it and others would say, "Ah! that's a lot." We get people calling just to find out how much it is before they can even come and among them are people who come to ask and never return. So, there is a possibility that the amount alone scares them away."* Radiographer, 6 years of experience

*"[. . .] If I have 4 patients, only 2 might come for the test and the rest will say that the cost is too high. It is not that the patients don't want to have the test, it is because it is very costly.–* Radiologist, 15 years of experience

**Sub-theme 2: Operational and technical barriers.**  *Mechanical breakdowns*. The participants stressed that mechanical failures of MRI machines often result in delays in dementia diagnoses. However, this concern was more pronounced among participants from public health facilities, where MRI machines may remain non-functional for extended periods, sometimes spanning months. They further highlighted that this issue is exacerbated by a shortage of readily available technical personnel and replacement components for the machines. The following quotes illustrate the participants' perspectives on mechanical breakdowns:

*"[. . .] We got a breakdown last year in February and it has happened again this February. Sometimes we get these issues and the problem is that we can't access the spare parts locally, they have to be imported which is a lengthy process and they are also expensive. . . During the COVID-19 pandemic, we had problems with traveling and it (the breakdown) took from March to August. It was around 6 months and it was because of transportation during that time. Usually, it takes a month or less than a month."*–Radiographer, 10 years of experience

*"[. . .] Yes, I guess maybe in Y Hospital. It (breakdown) is more in Y Hospital than in Z Hospital because the MRI in Y Hospital hasn't been working for a while and it only started working a few weeks ago. It happens especially in government hospitals. . . In Z (Hospital), the longest it can take is a week and in Y, it can take months."*–Psychiatrist, 8 years of experience.

*Interpretation difficulties*. Among participants affiliated with private-for-profit health facilities, there was a notable advantage in utilizing MRI machines equipped with pre-programmed protocols. However, radiologists operating in public health facilities acknowledged difficulties in interpreting specific MRI sequences and reports, subsequently impacting diagnostic accuracy. Moreover, some psychiatrists raised concerns about frequently receiving ambiguous reports from radiologists, leading to instances where clients have to incur additional costs due to the need to repeat those scans or get a second opinion, hence necessitating a new reading

from another radiologist. This difficulty in interpretation can be seen in the following responses:

*"[. . .] that's a problem that we have in all laboratories and imaging centers. They can't write imaging reports or are not informative about what you ask for. . . because MRI is new in Uganda, many multiple centers are doing them and they don't necessarily use very experienced people."*–Psychiatrist, 40 years of experience

*"[. . .] Also, there are deficiencies in terms of interpretations for MRI; If they just give you a slide or an image you may find it hard to read because even those that are supposed to help interpret are lacking in interpreting MRI scans."*- FGD with Radiology SHOs, Mak CHS.

*"[. . .] previously I have had two incidences and the quality is okay however, there are issues with interpretation. We have had issues where the interpretation from X hospital was actually wrong and we had to involve a senior who called the person who had done the MRI to revisit the MRI."*–FGD with psychiatrists, Mulago Hospital

**Sub-theme 3: Patient barriers.** *Claustrophobia*. The participants in this study shared that many patients suspected of dementia come with unmanageable fear for confined spaces resulting in physical movements during an MRI scan which distorts the accuracy of the readings. They stated that in some cases, some patients require sedation which is an extra cost to an already expensive service for them. Below are some of the quotes that bring out this theme:

*"[. . .] during the MRI, we need someone to be still and some dementia patients can't, so we have to sedate them so that we are able to have them in the MRI machine for 30 minutes without moving. . . It (sedation) is on the patient's cost because the anesthesiologist we use comes from an external facility like Mengo Hospital which causes extra cost to the patient. I know that patients won't be able to afford the sedation in the first place. . . it (sedation) is 200,000shs extra on top of the 550,000shs for the MRI. For some patients, we ask for 750,000shs because the doctor on site could think that the brain vessels need to be examined as well since there are cases of vascular dementia."*–Radiographer, 10 years of experience

"[. . .] People fear MRI that it will put you in a tunnel and some call it a grave or coffin but it has to be set up this way. So, it is really an educational challenge."–Radiologist, 30 years of experience

*"[. . .] First of all, there are attendants who are reluctant; the MRI machine is in an enclosed space and some attendants or their patients may find it uncomfortable going through with the MRI because it involves being in an enclosed space for a long time."*–Radiologist, 14 years of experience

*Dementia myths and misconceptions*. The participants emphasized that majority of the patients and caretakers do not have the right information about dementia which consequently affects the appropriate use of brain MRI in diagnosing dementia. They added that some patients also carry the wrong expectation of MRI findings which makes them feel cheated on seeing the scan results. They pointed out that the majority of caretakers of dementia patients do not know the right cause of their patient's symptoms leading to poor health seeking behaviors and late presentation for care. This is illustrated in the quotes below:

*"[. . .] I haven't met many of them but the majority are reluctant to come to the hospital because in most cases they don't understand what the disease is about. Most people think that*

*the disease can't be handled in the hospital even if they did come."–Radiographer, 6 years of experience*

*"[. . .] people expect to see a physical tumor to explain the dementia. So, when they see what the radiological image of dementia is, they are not convinced and they feel cheated. They expect to see more than the usual."–Psychiatrist,*

## Discussion

This study aimed at exploring the perceptions and practices of imaging personnel and physicians in using brain MRI as a diagnostic approach for dementia in Uganda.

Findings from the study demonstrated that there's a specialist gap for interpretation of brain MRI examinations related to dementia. The Radiologists do not have all the requisite competencies needed to correctly interpret the brain MRIs in patients with dementia. This is most likely due to the fact that MRI is a relatively new imaging modality hence there is limited exposure during training. This resonates well in many low resource settings where MRI is not relatively present in many health facilities. For example, it has been reported that less than one-third of the countries in Africa have one MR system serving less than a million population [21]. This means that many trained radiologists receive limited exposure to MRI which may consequently limit their ability to correctly interpret many MRI investigations. However, the life expectancy in Uganda where this study was done, has almost doubled hence the training of radiologists should now incorporate significant aspects of MRI imaging in dementia [9]. Although the situation is now improving with more MR systems installed, more exposure to MRI is still much needed. To address this, the training of radiologists now involves exposure to MRI training.

Findings from the study also showed limited access to MRI in the management of dementia because majority of hospitals do not have MRI equipment. This observation has been reported in other settings [21, 22]. The plausible explanation for this situation relates to the high costs involved to both the patients and the hospitals. The purchase and maintenance of MRI equipment is very expensive for many hospitals in low resource settings. Such expenses are transferred to the patients, majority of whom cannot afford. This thus means many of the dementia patients may not be able to undergo the examination due to the high costs involved as evidenced by the responses in this study. This finding quite resonates well with previously reported literature on MRI costs [21]. Relatedly, the majority of MRI scanners are centered within the capital city, hence majority of the dementia patients located in rural areas may not easily access MRI services. Consequently, this poses a challenge when brain MRI is adopted as a standard work up procedure for people with dementia. If majority of MRI machines are located within the urban centres, only a small part of the elderly population will be served [23]. Similarly, a study conducted in USA highlighted challenges such as limited skills in interpreting MRI results, the high cost of neuroimaging and turnaround time for MRI results [24].

Despite the challenges observed, responses from the participants showed a positive attitude towards using brain MRI for dementia diagnosis. All participants recognized the fact that MRI should be part of the management work up for patients with dementia.

A study by Boise on diagnosing Dementia, showed that some physicians only considered MRI if there was an unusually rapid progression of symptoms, a focus to the neurological symptoms when the patient was quite young. Whereas other physicians reported that one can't have a diagnosis of dementia without having had neuroimaging such as MRI [25] demonstrating a positive use of MRI in dementia diagnosis. Similarly, studies conducted in the developed world reported a positive use of MRI in the Diagnostic Evaluation of Patients with

Suspected Mild Cognitive Impairment and Alzheimer's Disease as well as positive perception and expectations of brain MRI use in improving patient care in dementia [26, 27]. Using Brain MRI to diagnose NDDs is a common standard in neurologic practice, and is a key element of differential diagnosis [28] and while there are some barriers to use such as the high costs, many specialists consider MRI the foundation for the accurate differential diagnosis among patients with dementia.

Despite the availability of MRI protocols for diagnosing dementia, their routine use is hindered by insufficient training in brain MRI specific to dementia. This lack of specialized training negatively impacts the diagnostic yield of MRI studies as healthcare providers may not fully utilize these protocols. Therefore, enhancing training programs focused on dementia-specific MRI protocols could potentially improve diagnostic accuracy and patient outcomes. The trend is now changing as dementia related aspects are now part of radiology training even in LMICs. This will ensure that imaging professionals will now have competencies to interpret brain MRIs in patients with dementia. In addition, using and adhering to standard brain MRI protocols in patients with dementia is crucial for early accurate diagnosis, treatment planning and monitoring disease progression. It also ensures consistent image acquisition as different protocols may induce systemic biases, allowing for reliable comparison between patients over time [29].

Participants identified claustrophobia and misconceptions about dementia as significant barriers to seeking diagnostic services, a finding corroborated by existing literature [30–33]. Therefore, it is crucial to enhance public awareness regarding the progression of dementia and to educate individuals about the diagnostic methods, including MRI imaging and other procedures, to facilitate early and accurate diagnosis.

From literature, majority of brain MRI studies in dementia have been conducted in high income settings. This study examines the practices, facilitators, and barriers associated with the use of brain MRI in diagnosing dementia. It adds to the global body of literature by presenting evidence on how imaging personnel and physicians perceive and utilize brain MRI for patients with suspected dementia in a resource-limited setting. Findings from this study have key implications for practice; first, MRI in dementia may not necessarily be rolled out at ago country wide due to the high costs involved, limited exposure of imaging personnel as well as equipment being only concentrated in the capital. Thus there is need for phased roll out of such practice as and when the MRI equipment becomes available country wide. In addition, targeted training initiatives including strengthening the residency training curriculum, formulating guidelines, and policy changes around the diagnostic testing procedures will enable more physicians to effectively order for structural brain MRI, so as to ensure improved diagnosis of dementia via neuroimaging.

Although this study was conducted in a single setting, which may not accurately represent other contexts with different policies or health systems, it nonetheless provides valuable information that can serve as a foundation for future research. Further research in this area is thus recommended especially from other low resource settings where brain MRI in dementia is yet to take root.

## Conclusion

This study highlights significant barriers to the effective use of brain MRI for diagnosing dementia, including limited specialist training, high costs, and uneven distribution of MRI equipment especially when it comes to low resource settings. Despite these challenges, there is a positive attitude among healthcare providers towards the adoption of MRI for dementia diagnosis. Enhancing training programs, improving public awareness, and implementing

phased rollouts of MRI services can improve diagnostic accuracy and patient outcomes. The findings will potentially help identify ways to scale up the use of brain MRI for dementia diagnosis, especially in low-resource settings, amid an increasing aging population. Future research should focus on similar low-resource settings to further validate these findings and develop comprehensive strategies for broader implementation.

## Supporting information

**S1 Data.**
(XLSX)

## Author Contributions

**Conceptualization:** Rita Nassanga, Noeline Nakasujja, Mark Kaddumukasa, Stephen E. Jones, Michael Grace Kawooya.

**Data curation:** Rita Nassanga.

**Formal analysis:** Rita Nassanga.

**Funding acquisition:** Rita Nassanga, Mark Kaddumukasa, Martha Sajatovic.

**Investigation:** Rita Nassanga.

**Methodology:** Rita Nassanga, Mark Kaddumukasa, Stephen E. Jones, Michael Grace Kawooya.

**Project administration:** Rita Nassanga, Mark Kaddumukasa.

**Resources:** Rita Nassanga.

**Supervision:** Mark Kaddumukasa, Stephen E. Jones, Martha Sajatovic, Michael Grace Kawooya.

**Validation:** Rita Nassanga.

**Visualization:** Rita Nassanga.

**Writing – original draft:** Rita Nassanga.

**Writing – review & editing:** Rita Nassanga, Noeline Nakasujja, Mark Kaddumukasa, Stephen E. Jones, Martha Sajatovic, Michael Grace Kawooya.

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
