## [Decision Letter · Decision Letter 0]

9 Jul 2024

PONE-D-24-21012Perceptions and Practices of Imaging Personnel and Physicians Regarding the Use of Brain MRI for Dementia Diagnosis in UgandaPLOS ONE

Dear Dr. Nassanga,

Thank you for submitting your manuscript to PLOS ONE. After careful consideration, we feel that it has merit but does not fully meet PLOS ONE’s publication criteria as it currently stands. Therefore, we invite you to submit a revised version of the manuscript that addresses the points raised during the review process.

**ACADEMIC EDITOR: The reviewers have given useful comments that will assist to further refine the paper. In addition to the reviewer comments, the authors need to strengthen the discussion by showing the relevance of this work to a global audience especially in LMICs where population demographics are changing and life expectancy is increasing yet use of MRI in the management of dementia has been less applied. Also, discuss the issue of limited training of the radiologists in use of MRI in dementia diagnosis. If they are not trained, this is likely to remain a challenge even after this study. In the methods section, the authors should show how rigor was ensured in the study since it was qualitative when it comes to issues of credibility, confirmability and dependability as well as transferability. How did the position of the researcher influence the research process? Is the researcher part of the people that were being studied? Issues of positionality have not been indicated in the methods. Lastly, please proof-read the entire paper to eliminate grammatical errors.**==============================

We look forward to receiving your revised manuscript.

Kind regards,

Aloysius Gonzaga Mubuuke

Academic Editor

PLOS ONE

 [The study took place in Uganda funded by the National Institutes of Health under the Brain Health Project].  

4. In the online submission form, you indicated that [(more...)

Data cannot be shared publicly because of restrictions from Ethics Committee. Data are available from the corresponding author on reasonable request.]. 

Additional Editor Comments:

The reviewers have given useful comments to guide the revision of the paper.

Reviewers' comments:

Reviewer's Responses to Questions

**Comments to the Author**

1. Is the manuscript technically sound, and do the data support the conclusions?

Reviewer #1: Partly

Reviewer #2: Yes

2. Has the statistical analysis been performed appropriately and rigorously? 

Reviewer #1: N/A

Reviewer #2: N/A

3. Have the authors made all data underlying the findings in their manuscript fully available?

Reviewer #1: No

Reviewer #2: Yes

4. Is the manuscript presented in an intelligible fashion and written in standard English?

Reviewer #1: No

Reviewer #2: Yes

5. Review Comments to the Author

Reviewer #1: The manuscript seems to describe a technically sound piece of scientific research with data that supports the conclusions. However, there are several clarifications and corrections that needs to be done to make the manuscript clearer and reflect your objectives.

Where as the author chose yes for the first question on data availability, no source of the data is provided instead the author assert that data cannot be shared publicly because of restrictions from Ethics Committee and that the data are available from the corresponding author on reasonable request.

There are several grammatical errors throughout the manuscript which can be dealt with at the rounds of review.

Reviewer #2: 1.The use of "Practices" was not well defined in this manuscript. Its use in the results section looks more of MRI utilization or uptake as opposed to the actual practice.

for example when you say Good practice, I would expect to see the appropriate application /use of MRI in the diagnosis of dementia; Poor practice may indicate some issues relatable to incompetency or in appropriate use of MRI or its abuse in the diagnosis of dementia. How ever what is indicated in Domain A of the results section are more of the influencers of MRI utilization or uptake; refer to the Domain 1,theme 3;Poor Practice (low patient numbers, cost of MRI) the attributable factors are more of the barriers mentioned in Domain C.

2.There is need to have a consistent format in the quotations; i.e some quotes indicate participants age and experience well as other quotes only indicate experience.

6. PLOS authors have the option to publish the peer review history of their article (what does this mean?). If published, this will include your full peer review and any attached files.

Reviewer #1: No

Reviewer #2: **Yes: **Dr.Tumwine Moreen.

---

## [Author Response · Author response to Decision Letter 0]

11 Aug 2024

Rita Nassanga

Makerere University, College of Health Sciences

Department of Radiology and Radiotherapy 

Date:19th. July.2024

The Academic editor 

Plos-one

Dear Sir, 

RE: Response to the reviewer’s comments

The issues raised by the editor have been addressed in the revised paper as summarized in the table below. 

Serial # Comment Response 

 ACADEMIC EDITOR: The reviewers have given useful comments that will assist to further refine the paper. In addition to the reviewer comments, the authors need to strengthen the discussion by showing the relevance of this work to a global audience especially in LMICs where population demographics are changing and life expectancy is increasing yet use of MRI in the management of dementia has been less applied. Also, discuss the issue of limited training of the radiologists in use of MRI in dementia diagnosis. If they are not trained, this is likely to remain a challenge even after this study. In the methods section, the authors should show how rigor was ensured in the study since it was qualitative when it comes to issues of credibility, confirmability and dependability as well as transferability. How did the position of the researcher influence the research process? Is the researcher part of the people that were being studied? Issues of positionality have not been indicated in the methods. Lastly, please proof-read the entire paper to eliminate grammatical errors.

 The discussion has been strengthened to show relevance of this work to a global audience. 

The issues of rigor, credibility, confirmability and dependability as well as transferability have been addressed in the methods section. 

 ABSTRACT 

1. Abstract-Introduction: The statement “In Uganda, MRI use for dementia diagnosis is infrequent” should be re-written as, “the use of MRI in diagnosis of dementia is infrequent in Uganda.” The statement “and subtle findings like mild regional atrophy are often overlooked, despite being crucial for imaging diagnosis” should be revised to make a clear meaning that connect well with the preceding statement. E.g., you could have said, “Even when it is used, subtle findings like mild regional atrophy are often overlooked, despite being crucial for imaging diagnosis.” Please revise and clarify the introduction in your own words.

 Edited as suggested, in the introduction section of the abstract

2. Abstract-Methods; write the method clearly to include the study design, study setting, study participant, sampling method/sample size, data collection method, and data analysis. These should be short, clear and concise statements.

 Edited as suggested, in the methods section of the abstract. 

3. Abstract-Results: remove the number of participant from the result section. 

In qualitative research abstract, you could summarize your result by presenting the themes and the key message from the themes.

 Edited as suggested; the number of participants has been moved to the methods section of the abstract. 

The themes, sub-themes and key messages have been stated in the results section of the abstract. 

 INTRODUCTION 

4. You need to re-write and reorganize the introduction for clarity as well as improve the grammar. I have made a few but not all the suggestions to help you out.

 The introduction has been re-written and organized as suggested. 

5. There is a lot of mix up of sentences; in paragraph 2 you are talking about situation in Uganda, then you brought definition and global perspective in paragraph three. Then in paragraph 4 you jumped to LMIC and back to Uganda in paragraph five. Is better to start with your brief opening statement, define, then global literatures/figures followed by LMIC and finally organize the local context/Ugandan literature in one paragraph

 The introduction has been re-arranged; the order is;

Definition of dementia, llobal literature, LMIC literature and lastly the Ugandan literature. 

6.. Your definition of dementia should ideally come earlier than in paragraph 3. 

 The definition of dementia has been moved to paragraph one. 

7. Where as in paragraph one you seem to imply that more than half of dementia is in LMIC, you invalidated this statement in paragraph 2 

 This has been corrected. 

8. There are statements in the last paragraph that needs referencing. Otherwise you are making unjustified claims.

 Two references from studies done in Uganda have been added. 

9. The last statement “the findings from the study will potentially help to identify ways of scaling up the use of brain MRI in the diagnosis of dementia especially in low resource settings in the wake of the increasing aging population.” is inappropriate in the introduction section and could appear as part of your conclusion or recommendation. It implies that you had pre-conceived result of your study prior to conducting it. 

 This statement has been moved to the conclusion section. 

10. Your introduction does not shade light into the practices of MRI use in dementia diagnosis globally, regionally or in the local context. You need to present what is known about your study topic (practice and perception) in global perspective or developed countries or LMIC. Otherwise this would imply that you are doing a ground breaking research.

 Previous studies on the Practices and perceptions on the use of brain MRI in the diagnosis of dementia, globally and locally, have been cited. 

11. What is the current guideline or standard in dx of dementia whether in Uganda or in other settings

 The standard of diagnosis in dementia in Uganda and other settings has been stated in paragraph four of the introduction. 

12. What are the perceptions on the use of MRI in other setting?

 Previous studies on the Practices and perceptions on the use of brain MRI in the diagnosis of dementia, globally and locally, have been cited.

13. In summary, your introduction should make it clear as to where your research sits within the wider scholarly network, what is known about the topic, what the unknown is, and which gap in knowledge is it addressing. 

 Further clarification has been provided in the introduction and all changes have been incorporated as suggested. 

 METHODOLOGY 

14. Your methodology requires thorough revision. The sentences are poorly constructed with several grammatical errors and mix-up. I have a few questions and suggestions for clarifications and corrections.

 Grammatical errors have been rectified.

The mix-up has been clarified. 

15. You have confused me with your study design. Please first state the study design, then in another paragraph you can write about the study setting and finally the data collection method. Otherwise you have mixed up the write up. Look at this example “This was an exploratory qualitative study. The study was conducted in……”

 The introductory part of the methodology has been clearly broken down into;

Study design

Study setting

16. Elaborate on your study setting; where in Kampala was the study conducted and what is the characteristic of your study setting.

 The study took place at seven urban based health facilities in Kampala, Uganda, and each had a 1.5T MRI scanner. This has been clarified in the study setting

17. You mixed up your study settings and study participants. You are talking about the nature of MRI scanner in some facility as if those are part of the study participants. The study setting has been re-written and the participants have been moved to the participants and sampling section of the methodology. The nature of the MRI scanner has been moved to the study setting. 

18. You used diverse category of participants. How did you sample the different groups of participant? How did you determine the sample size for each category of participants? All use MRI in several ways; clinicians refer, imaging techs carry out the MRIs whereas the Rads interpret these MRIs; we wanted to get a variety of responses from all stake holders /health professionals to maximum variation. 

19. Comment on your inclusion and exclusion criteria for each category of participants

 The inclusion and exclusion criteria for each category of participants has been stipulated in the methodology. 

20. How where the study participants recruited? This being a purely qualitative study, the participants were purposively selected on basis of their experience with brain MRI or diagnosis of dementia. The information has been included in the the participants and sampling section of the methodology.

21. In which respective work place was the FGD conducted? You didn’t specify the work place of the study participant.

 The two FGDs were conducted in of the healthy facilities; the name of the specific facility has been left out due to ethical reasons of confidentiality. 

22. Did you use a validated data collection tool? If not, how did ensure credibility of the data collected. Did you use the same data collection tool for all the different category of participant?

 The questions for the FGDs and KIIs were informed by literature and first pre-tested with four experts to ensure their credibility.

23. How did you determine your saturation point and at what point did you achieve saturation for FGD and KIIs.

 Saturation was achieved at the 10th interview however, one more was included to confirm saturation; this has been included in the revised paper. 

24. You did only two FGDs involving two totally different category of participant. Please comment of data saturation for FGD and how you analyzed these two pieces of data. 

 These are two trainee groups involved in the assessment of dementia patients. However, carrying out only two FGDs could be a limitation of the study and has been included as part of the limitations in the discussion. 

25. How did you analyze the data from the different categories of participants? Did you analyze them as aggregated data? 

 Yes the data was analysed as a whole because the FGD and KII questions were related. 

26. Referencing in methodology is inappropriate. It is implied that your method is based on well-established methodology. The steps that were followed were not for the researchers but were borrowed from previously documented literature. 

 RESULTS 

27. In your table of results you presented domains and themes yet in your data management and analysis you mentioned having generated codes, sub-themes and themes.

 A detailed table that shows the codes, themes, sub-themes and illustrative quotes has been included in the results section. 

28. You need to thoroughly improve the presentation of your results. Make your sentences clearer and correct use of grammar. You may need to re-visit your themes/codes and quotes again

 We have revisited the raw data and the quotes as well as the themes have been verified. We have also included a new detailed table showing the themes, sub-themes, codes and related quotes. 

29. You had two objectives i.e., exploring perception and practice but your result does not to make it clearer for the reader how you addressed both objectives. I am interested in your results on the practice and perceptions as separate findings from the study.

 The analysis followed an inductive approach where themes and sub-themes purely emerged from the data therefore, the perceptions and practices of the participants were indirectly expressed through the presented themes and sub-themes as seen in table 1. 

30. The entire paper needs to be revised thoroughly for grammatical errors and clarity.

31. The manuscript seems to describe a technically sound piece of scientific research with data that supports the conclusions. However, there are several clarifications and corrections that needs to be done to make the manuscript clearer and reflect your objectives.

Whereas the author chose yes for the first question on data availability, no source of the data is provided instead the author assert that data cannot be shared publicly because of restrictions from Ethics Committee and that the data are available from the corresponding author on reasonable request.

There are several grammatical errors throughout the manuscript which can be dealt with at the rounds of review.

 The whole paper has been revised and clarifications have been made. 

Raw data will be available on individual request form the corresponding author due to IRB restrictions on data sharing. 

The entire paper has been proof-read to correct the grammatical errors. 

32. The use of "Practices" was not well defined in this manuscript. Its use in the results section looks more of MRI utilization or uptake as opposed to the actual practice.

for example when you say Good practice, I would expect to see the appropriate application /use of MRI in the diagnosis of dementia; Poor practice may indicate some issues relatable to incompetency or in appropriate use of MRI or its abuse in the diagnosis of dementia. However what is indicated in Domain A of the results section are more of the influencers of MRI utilization or uptake; refer to the Domain 1,theme 3;Poor Practice (low patient numbers, cost of MRI) the attributable factors are more of the barriers mentioned in Domain C.

 In the context of the study, practices referred to uptake or utilization of MRI.

33. There is need to have a consistent format in the quotations; i.e some quotes indicate participants age and experience well as other quotes only indicate experience.

 The quotes have been edited for uniformity. 

Yours Sincerely,

Dr. Rita Nassanga 

Corresponding author

Department of Radiology and Radiotherapy

College of Health Sciences, Makerere University.

---

## [Decision Letter · Decision Letter 1]

30 Aug 2024

PONE-D-24-21012R1Perceptions and Practices of Imaging Personnel and Physicians Regarding the Use of Brain MRI for Dementia Diagnosis in UgandaPLOS ONE

Dear Dr. Nassanga,

Thank you for submitting your manuscript to PLOS ONE. After careful consideration, we feel that it has merit but does not fully meet PLOS ONE’s publication criteria as it currently stands. Therefore, we invite you to submit a revised version of the manuscript that addresses the points raised during the review process.

**ACADEMIC EDITOR: **The revised paper reads better though it still needs some refinement. In addition to the reviewer comments raised, please proof-read the paper to ensure that it flows well and correct the occasional grammatical errors. ==============================

We look forward to receiving your revised manuscript.

Kind regards,

Aloysius Gonzaga Mubuuke

Academic Editor

PLOS ONE

Additional Editor Comments:

The revised paper reads better though it still needs some refinement. In addition to the reviewer comments raised, please proof-read the paper to ensure that it flows well and correct the occasional grammatical errors.

Reviewers' comments:

Reviewer's Responses to Questions

**Comments to the Author**

1. If the authors have adequately addressed your comments raised in a previous round of review and you feel that this manuscript is now acceptable for publication, you may indicate that here to bypass the “Comments to the Author” section, enter your conflict of interest statement in the “Confidential to Editor” section, and submit your "Accept" recommendation.

Reviewer #1: (No Response)

2. Is the manuscript technically sound, and do the data support the conclusions?

Reviewer #1: Yes

3. Has the statistical analysis been performed appropriately and rigorously? 

Reviewer #1: N/A

4. Have the authors made all data underlying the findings in their manuscript fully available?

Reviewer #1: Yes

5. Is the manuscript presented in an intelligible fashion and written in standard English?

Reviewer #1: No

6. Review Comments to the Author

Reviewer #1: The author partially addressed the concerns raised in the previous review. There is need for further thorough review before this work can be accepted.

7. PLOS authors have the option to publish the peer review history of their article (what does this mean?). If published, this will include your full peer review and any attached files.

Reviewer #1: No

---

## [Author Response · Author response to Decision Letter 1]

5 Sep 2024

Rita Nassanga

Makerere University, College of Health Sciences

Department of Radiology and Radiotherapy 

Date: 02. Sept. 2024

The Academic editor 

Plos-one

Dear Sir, 

RE: Response to the reviewer’s comments

The issues raised by the editor have been addressed in the revised paper as summarized in the table below. 

Serial # Comment Response 

 ACADEMIC EDITOR 

 The revised paper reads better though it still needs some refinement. In addition to the reviewer comments raised, please proof-read the paper to ensure that it flows well and correct the occasional grammatical errors. The revised paper has been proofread after incorporating the reviewer comments. 

1. One of the main problems in the paper is the poor organization of your write up, lack of clarity and a lot of grammatical errors. Work on the grammar thoroughly and improve clarity of your write up. You seem not to be paying careful attention to clear connection between sentences and paragraphs, grammar, and clarity.

 The revised manuscript has been improved and proof read to correct the flow, grammar and clarity. 

 ABSTRACT 

2. Introduction: The introduction should just be a snap shot and well aligned but not too verbose. If I delete most of your English, I end up with this short introduction. Think of how you can write a clear and concise introduction that does not obscure the main thing you want to present.

“Introduction: Diagnosing dementia remains challenging in low-income settings due to limited diagnostic options. Though brain MRI is crucial in dementia diagnosis, its use in Uganda remains infrequent.”

 This has been revised as suggested. 

3. Methods: Change “and analyzed thematically using an inductive approach” to “and analyzed by inductive thematic analysis.” 

 This has been edited as suggested. 

4. Results: You don’t need to list your subtheme here. After listing the themes, just present the key message from the result as the next statements. Please remove the subthemes and refines the remaining write up for clarity.

 The subthemes have been removed as suggested. 

5. Conclusions: your write up lacks proper coordination; re-write the conclusion and make the take home message very clear. How do you convince someone about adopting MRI use when your take home message is casting only doubt by presenting mainly draw-backs. You need to write and also provide positive side of your finding to convince the reader why MRI is worth adopting despite the barriers.

 The conclusion was edited as advised. 

 MANUSCRIPT BODY 

6. Your introduction is still so mixed up; you need to organize your argument into coherent write ups which connect well with each other. Please note that writing a compelling introduction/background is a sort of storytelling. There should be clear connection and flow of message from one paragraph to the next. 

Your paragraph one could only focus on the definition and type/cause of dementia you stated. Please remove “it affects 7–8% of individuals older than 65 years old and 30% of individuals above 80 years old [8]” and find were to fit it under epidemiology/risk factors in paragraph 2

 The entire introduction has been revised as suggested to make it flow as suggested. It has also been proof-read to correct the grammar. 

7. Then organize the paragraphs on epidemiology/prevalence and risk factor for dementia and connect the narrative well. I can’t find the clear flow in your paragraph two; you started globally, and then mention Africa, before jumping to LMIC then back to Africa/sub-Saharan Africa again. You could first present the global burden, then LMIC then Africa/sub-Saharan and finally the prevalence in Uganda. You can then bring in the drivers of dementia; you mentioned age as the major driver, but then connect it well with life expectancy in driving aging population. Then you can end the paragraph with the concerns related to the rising prevalence you tried to state.

 The paragraphs have been reorganized as suggested. 

8. Connect the third paragraph to the second using preposition such as despite, inspite of, although, etc. Example of a good start is “despite the rising prevalence of dementia, diagnosis is still very challenging…………………………………………………………………………….…………..”

 This has been done. 

9. The statement “model paper[14]In a previous study on perceptions on dementia diagnosis, it was reported that” doesn’t sit well. The sentence following it connects better to the preceding paragraph. You either delete that statement of find better way of putting it in that story you are trying to tell.

 This has been deleted. 

10. I am wondering how the lack of a diagnostic tool results in high prevalence of dementia in Uganda. Does that mean clinicians in Uganda are over diagnosing dementia? Please clarify on that statement and how that is possible, in way that a reader can get convinced.

 This has been reworded and prevalence of dementia has been replaced with challenges in diagnosing dementia. 

11. The last paragraph is good but you need to improve it. The sentences are too long; it also seems like you miss to put a full stop after the first sentence. 

 The sentences have been summarized and the paragraph has been improved. 

 METHODS 

12. Study design; Remove the statement “conducted in health facilities with MRIs” because that appears in your study setting.

 This has been done. 

 13. You don’t necessary to need to write inclusion and exclusion criteria as sub-headings. It should be seen in your description of the study participant. 

 The inclusion and exclusion criteria sub-headings have been removed. 

14. What you have put as an exclusion criterion may not be correct for this study methodology. You exclude those who participated in the study. E.g. if for any reason you can’t analyze results from certain participant then you exclude them and give the reason for the exclusion. Similarly, you didn’t use convenience sampling. Therefore the absence of a participant can’t be an exclusion criterion for purposive sampling 

 This has been removed. 

15. How long were the KII and FGDs? Each KII lasted forty minutes whereas ech FGD lasted one hour. 

16. Am still wondering how you used the same interview guide for technologist, radiologist psychiatrist, and Neurologist. These cadres interact with the brain MRI at different level. But if you were able to get what you set out to do then well and good. The questions in the interview guide were formulated ton apply to all participant cadres because they were exploring their experience in general. 

 RESULTS 

17. You may not need to put quotes in your table of result. It becomes a repetition as the same quotes appear down in the results below the table unless you are presenting the results in its entirety in a tabular form. 

 Not all quotes in the table are repeated in the description of the themes. The quotes in the narrative are just a small representation of the quotes in the table just to relate to each theme. The quotes in the table are meant to show the reader at a glance illustrative responses against each theme. In the last review comments, one of the reviewer srecommended to include some quotes in the table.

18. In your methodology, this is the participant categories you provided; the participants comprised three radiologists, six Radiology residents, eight Psychiatry residents, two psychiatrists, two neurologists, and four imaging technologists. However, the demographic categories presented in the result section are different. You now changed to “the sample included ten psychiatrists, two neurologists, three radiologists, and ten radiographers.” Such inconsistency is not good; you should be more alert in your writing to avoid such discrepancies and confusion. 

 This has been corrected in the results section to tally with the methodology. 

 DISCUSSION 

19. Your discussions are also mixed up; the essence of discussion is to make the message from your finding clear and provide implication of your findings to practice.

 In paragraph one you discussed limited skills in MRI interpretation. Paragraph two you discussed limited MRI access but in the last sentence you concluded with this statement “Similarly, a study conducted in USA highlighted challenges such as limited skills in interpreting MRI results, the high cost of neuroimaging and turnaround time for MRI results.” Stick to the main theme of the discussion in each paragraph, you have already discussed limited skills in the preceding paragraph.

 This section has been revised to focus on skills only. 

20. Looking at this part of your discussion, “A study by Boise on diagnosing Dementia, showed that some physicians only considered MRI if there was an unusually rapid progression of symptoms, a focus to the neurological symptoms when the patient was quite young. Whereas other physicians reported that one can't have a diagnosis of dementia without having had neuroimaging such as MRI [25] demonstrating a positive use of MRI in dementia diagnosis.” Put it in context of your findings rather than copying text. Try to re-write to bring the aspect of how participant in your study perceive prioritizing MRI. How they select patients for MRI investigation given the financial constraint. Bring clearly the key message of the participant. Do the same throughout your discussion, and if you have any key recommendation or gap based on your study, make them along. 

 This section has been revised to incorporate the changes as suggested. 

21. Paragraph three, you are again discussing “insufficient training in brain MRI” which actually appeared in paragraph 1. Please merge the two discussions into one meaningful write up.

 The two paragraphs on limited skills and access have been merged.

22. The statement; “participants identified claustrophobia and misconceptions about dementia as significant barriers to seeking diagnostic services, a finding corroborated by existing literature [30-33].” Please cite examples of the misconception from your finding before you continue the discussion. Whose misconceptions are being discussed here? Is it the public or practitioners’ misconception?

 The misconceptions being referred to are of the patients. This has been clarified. 

23. Give feasible recommendations that can improve practice and diagnosis of dementia in Uganda. Because based on your findings, the cost of MRI is the main prohibition of its use especially in the poor communities. I am wondering “what you mean by a phase rollout of MRI services” and how such a roll out will mitigate the financial constraints or increase patient’s capacity to afford brain MRI. 

 The recommendations have been revised. 

24. Remember I asked you previously on how dementia is being diagnosed in Uganda (the current practice in diagnosis of dementia). True MRI is valuable and superior, but is only in Uganda Capital according to your literature. What do you recommend in the event that cost and limited access is involved and dementia patient are there in the communities not being diagnosed? 

 This statement has been included; In the beginning, the few available MRI centres need to be fully utilized by encouraging referral of dementia patients to these centres and coming up with a process of subsidizing costs for them.

 Conclusion 

25. Work on grammar and clarity. Provide concise, clear conclusions based on your findings. Remember you are making your case summary to convince someone on MRI adoption. How do you convince someone about adopting MRI use when your take home message is casting doubt by only presenting draw-back?

 The conclusion has been revised to highlight the positive use of MRI in dementia despite the challenges. 

Yours Sincerely,

Dr. Rita Nassanga 

Corresponding author

Department of Radiology and Radiotherapy

College of Health Sciences, Makerere University.

---

## [Editor Report · Decision Letter 2]

11 Sep 2024

Perceptions and Practices of Imaging Personnel and Physicians Regarding the Use of Brain MRI for Dementia Diagnosis in Uganda

PONE-D-24-21012R2

Dear Dr. Nassanga,

We’re pleased to inform you that your manuscript has been judged scientifically suitable for publication and will be formally accepted for publication once it meets all outstanding technical requirements.

Kind regards,

Aloysius Gonzaga Mubuuke

Academic Editor

PLOS ONE
---

## [Editor Report · Acceptance letter]

21 Sep 2024

PONE-D-24-21012R2 

PLOS ONE

Dear Dr. Nassanga, 

I'm pleased to inform you that your manuscript has been deemed suitable for publication in PLOS ONE. Congratulations! Your manuscript is now being handed over to our production team.

Kind regards, 

on behalf of

Dr. Aloysius Gonzaga Mubuuke 

Academic Editor

PLOS ONE